# Interleukin-31 Receptor A Expression in the Dorsal Root Ganglion of Mice with Atopic Dermatitis

**DOI:** 10.3390/ijms24021047

**Published:** 2023-01-05

**Authors:** Iwao Arai, Saburo Saito

**Affiliations:** Division of Environmental Allergy, The Jikei University School of Medicine, 3-25-8 Nishi-Shinbashi, Tokyo 105–8461, Japan

**Keywords:** alloknesis, atopic dermatitis, interleukin-31, interleukin-31 receptor A, itching, scratching

## Abstract

Atopic dermatitis (AD) is a common skin disease caused by genetic and environmental factors. However, the mechanisms underlying AD development remain unclear. In this study, we examined the genetic factors contributing to the onset of itch-associated scratching in different strains of mice. Interleukin-31 (IL-31) induces severe scratching and dermatitis in mice. However, the site of action of IL-31 remains unclear. Cutaneous IL-31 and IL-31 receptor A (IL-31RA) mRNAs in the dorsal root ganglion (DRG) are expressed exclusively in the AD model, i.e., NC/Nga mice. Here we evaluated the effects of repeated administration of IL-31 on the scratching behavior in NC/Nga, BALB/c, and C57BL/6 mice. The results showed that repeated administration of IL-31 significantly increased itch-associated scratching (LLS) behavior in the three strains of mice. One hour after an intravenous IL-31 injection, BALB/c mice showed alloknesis-like behavior. Mite infestation and IL-31 administration triggered itchy skin, increased LLS counts and DRG neuronal IL-31RA expression, and eventually caused dermatitis. The dermatitis severity and LLS counts induced by mite infestation and IL-31 administration were in the order NC/Nga > BALB/c > C57BL/6. In conclusion, neuronal IL-31RA expression in the DRG was the most important genetic factor affecting the severity of LLS and dermatitis in mice.

## 1. Introduction

Atopic dermatitis (AD) model NC/Nga mice with chronic itching spontaneously develop skin lesions, accompanied by immunoglobuline E (IgE) overproduction and skin inflammation, under conventional conditions but not under specific pathogen-free (SPF) conditions [1]. Given that itching elicits a strong desire to scratch, scratching behavior count is a useful index to evaluate itching [2]. Scratching behavior in mice is of two types: long-lasting scratching (LLS: scratching behavior lasting more than 1.5 s) and short-lasting scratching (SLS: scratching behavior lasting from 0.3 to 1.5 s). In our previous study on the scratching behavior of NC/Nga mice, we found that LLS is frequently observed in skin-lesioned NC/Nga mice, whereas SLS is frequently observed in skin-lesioned and not skin-lesioned NC/Nga and other strains of mice [3,4]. These results suggest that SLS is related to hygiene behavior, whereas LLS is the true itching response in these mice.

Increased LLS counts and dermatitis can be induced in various strains of mice cohoused with skin-lesioned NC/Nga mice [5]. However, these phenomena can be blocked by insecticidal pretreatment. Thus, mite infestation is responsible for the increased LLS counts and dermatitis after cohousing with skin-lesioned NC/Nga mice [6,7]. In addition, dermatitis does not develop under SPF conditions in NC/Nga mice.

Interleukin-31 (IL-31) is a T-cell cytokine whose overexpression causes pruritus and dermatitis similar to AD. IL-31 increases scratching and induces severe dermatitis in mice [8]. IL-31 signals through a heterodimeric receptor consisting of IL-31RA and oncostatin M receptor β (OSMR), and it has been linked to AD development [8]. IL-31 and its receptors IL-31RA and OSMR are involved in AD, pruritus, and dermatitis at the mRNA level [9,10]. The expression of IL-31 transcripts is significantly increased in skin-lesioned NC/Nga mice (Figure 1a) [11], and these changes coincide with increases in LLS counts [12]. IL-31 expression is involved in inflammation and itching in patients with AD. Serum IL-31 level correlates with serum IgE, eosinophil cationic protein, disease severity, and itch intensity in patients with AD [13,14,15]. Furthermore, IL-31 expression is upregulated in the lesioned skin of patients with AD [16]. Repeated administration of IL-31 significantly increases LLS counts and upregulates dorsal root ganglion (DRG) IL-31RA expression, and LLS counts and DRG IL-31RA expression are closely correlated [17]. Our recent finding showed that IL-31 is an important factor in inducing itching and promoting scratching in NC/Nga mice; however, the sites of action of IL-31 remain unclear.

Touch- or brush-evoked pruritus around an itching site is termed “itchy skin” or “alloknesis,” whereas pin-prick-evoked itching sensations around an itching site are termed “hyperknesis” [18]. Patients with chronic itching may perceive nonpruritic stimuli, such as electrical, noxious heat pain, or scratching stimuli distal to itchy stimuli, as itchy [19,20]. On the other hand, we measured the LLS counts before and after mild (10 times) mechanical scratching using by stainless-steel wire brush in skin-lesioned NC/Nga mice. We found that the LLS count increases immediately after the loading of mechanical scratching [21]. This result suggested that alloknesis-like phenomena occur in skin-lesioned NC/Nga mice [22]. Therefore, in the present study, we investigated itching in response to several pruritogens or algogens in BALB/c mice cohoused with skin-lesioned NC/Nga mice or intravenously administered with IL-31.

IL-31RA is a type I cytokine receptor that mediates IL-31 signaling when coupled with OSMR [8] and is expressed by several cell types, including monocytes, epithelial cells, T cells, and sensory nerves. Given that IL-31RA and OSMR are expressed in afferent cutaneous nerve fibers/DRG, IL-31 may link the immune and nervous systems. IL-31 is a helical cytokine belonging to the glycoprotein 130 (gp130)/IL-6 cytokine family, which also includes IL-6, leukemia inhibitory factor (LIF), oncostatin M (OSM), and neuropoietin [23]. All members of this family share the common chain of gp130 in their multi-unit receptor complexes, except for IL-31, which uses a gp130-like receptor [24], and are involved in various fundamental physiological processes, such as neuronal growth, bone metabolism, cardiac development, and immune regulation [25]. Notably, IL-31RA transcripts are abundantly expressed in the DRG of 63 different human tissues, where the cell bodies of primary sensory neurons are present [26,27].

We also examine the relationship between LLS and the expression of several mRNAs at different sites of action on sensory nerves in NC/Nga and BALB/c mice, in which LLS was induced by mite infestation or IL-31 administration, and to elucidate the regulatory mechanisms underlying itch development. Further, the relationship between itching and the expression of neuronal DRG IL-31RA mRNAs in NC/Nga, BALB/c, and C57BL/6 mice was analyzed, and the regulatory mechanisms underlying dermatitis development were elucidated. This study shows a relationship between IL-31, alloknesis, DRG neuronal IL-31RA expression, neuronal IL-31RA, LLS counts, and dermatitis severity.

## 2. Results

### 2.1. LLS Caused by Cohousing with Skin-Lesioned NC/Nga Mice

Mite infestation increased the LLS counts in SPF-NC/Nga mice cohoused with skin-lesioned NC/Nga mice for 3 days compared with non-cohoused (naïve) mice. This increase in LLS (Figure 1b) showed a circadian rhythm; in particular, the increase in LLS counts was significant during nighttime, whereas that during daytime was not significant (Figure 1d, shaded area: dark phase). The increase in SLS (Figure 1c) also showed a circadian rhythm and increased remarkably at nighttime, depending on the motor activity of the mice. However, no significant difference in SLS counts was observed between naïve and mite-infested mice (data not shown).

### 2.2. Time-Course Changes in Several Parameters of Dermatopathy in SPF-NC/Nga Mice Cohoused with Skin-Lesioned NC/Nga Mice

The LLS counts in the SPF-NC/Nga mice significantly increased 1–14 days after cohousing with skin-lesioned NC/Nga mice in a time-dependent manner (Figure 2a). The dermatitis score also increased 14 days after cohousing (Figure 2b). The mRNA expression levels of cutaneous IL-31 and IL-31RA increased during the experimental period, but the increase was temporary and elusive (Figure 2c,d). In contrast, DRG IL-31RA mRNA expression significantly increased 3–14 days after cohousing in a time-dependent manner (Figure 2e). LLS counts were closely correlated with DRG neuronal IL-31RA mRNA expression (Figure 2f, r = 0.861, *p* < 0.05) but not with cutaneous IL-31 (r = 0.302) and IL-31RA mRNA expression (r = 0.223).

### 2.3. Effects of Pruritogens and Algogens on LLS Counts in BALB/c Mice Cohoused with Skin-Lesioned NC/Nga Mice

Cohousing of BALB/c mice with skin-lesioned NC/Nga mice for 6 days caused itchy skin. Figure 3 shows saline-, acetylcholine-, bradykinin-, and serotonin-induced scratching behavior in the mice with itchy skin. These stimulants elicited many SLS counts, and few LLS counts in the naïve mice (Figure 3a). In contrast, the LLS counts were increased in the mice with itchy skin (Figure 3b). In the naïve mice, saline and algogens, such as acetylcholine, bradykinin, and capsaicin, elicited SLS but not LLS (Figure 3c, blue column). In addition, pruritogen, saline, histamine, serotonin (5-HT), and compound 48/80 elicited SLS but not LLS in the naïve mice (Figure 3d, blue column). These pruritogens elicited scratching 5 min after injection, and this behavior persisted for at least 30 min. The cohousing caused itchy skin, and pruritogens- and algogens-induced LLS counts were significantly increased in the cohoused mice compared with the naïve mice (Figure 3c,d, red column).

### 2.4. Effects of Cohousing with Skin-Lesioned NC/Nga Mice on LLS Counts in NC/Nga, BALB/c, and C57BL/6 Mice

The LLS counts in the SPF-NC/Nga mice rapidly increased from 17.3 ± 3.1 counts/24 h to 412.7 ± 42.9 and 984.6 ± 113.8 counts/24 h at 1 and 6 days after cohousing with skin-lesioned NC/Nga mice, respectively (Figure 4a). The LLS counts in the BALB/c mice gradually increased from 60.3 ± 9.6 counts/24 h to 141.6 ± 18.9 and 364.8 ± 32.0 counts/24 h at 1 and 6 days after cohousing with skin-lesioned NC/Nga mice, respectively (Figure 4b). The LLS reached a steady state with approximately half the counts in the NC/Nga mice (Figure 4a,b, red line). The C57BL/6 mice showed no significant changes in LLS counts, which were 58.3 ± 6.2, 79.0 ± 6.5, and 51.0 ± 12.1 counts/24 h at 0, 1, and 6 days after cohousing, respectively (Figure 4c).

### 2.5. Effects of Repeated Administration of IL-31 on LLS Counts and DRG Neuronal IL-31RA mRNA Expression in SPF-NC/Nga Mice

Repeated intradermal (i.d.) injection of IL-31 (1 μg/site) every 12 h for 3 days gradually increased LLS counts, which increased intermittently after each dosage of IL-31. This increase in LLS counts showed a circadian rhythm; in particular, the LLS counts significantly increased at nighttime, and itchy skin also increased after cohousing with skin-lesioned NC/Nga mice (Figure 5a, shaded area). The LLS counts after the first dose of IL-31 did not significantly increase, but those after the last dose were twice those after the first dose. They gradually increased from day 1 and reached a plateau 3 days after administration (Figure 5b). Repeated administration of IL-31 also increased the expression of DRG neuronal IL-31RA (Figure 5c) and OSMR mRNAs but not the expression of DRG neuronal LIFR or gp130 and cutaneous IL-31RA mRNA (data not shown) [17]. Cutaneous IL-31 and IL-31RA mRNA were not expressed during the experiment. The expression of DRG neuronal IL-31RA mRNA was closely correlated with the LLS counts (Figure 5d, r = 0.978, *p* < 0.05). These changes and the itchy skin were similar to cohousing with skin-lesioned NC/Nga mice.

### 2.6. Effect of IL-31 Pretreatment on Pruritogen- or Algogen-Induced Scratching Behavior in BALB Mice

Figure 6 shows traces of saline-, acetylcholine-, bradykinin-, and serotonin-induced scratching behavior 1 h after the intravenous injection of IL-31 in BALB/c mice. These stimulants elicited many SLS counts, and few LLS counts in the vehicle-treated mice (Figure 6a). In contrast, IL-31 pretreatment increased the LLS counts in the mice (Figure 6b). In the vehicle-treated mice, saline; algogens such as acetylcholine, bradykinin, and capsaicin; pruritogens such as histamine, serotonin; and compound 48/80 elicited SLS but not LLS (Figure 3c,d, blue column). At 1 h after IL-31 intravenous injection (50 μg/kg, i.v.), these LLS counts significantly increased as compared with those in the vehicle (phosphate buffer solution, PBS, 10 mL/kg) administered mice (Figure 6c,d, red column). These stimulants elicited scratching 5 min after the injection, and this behavior persisted for at least 30 min.

### 2.7. Effects of Repeated Administration of IL-31 on LLS Counts in NC/Nga, BALB/c, C57BL/6, and IL-31RAKI (IL-31RA-Deficient) Mice

Subcutaneous administration of IL-31 (50 μg/kg, s.c.) every 12 h for 3 days increased the LLS counts. The LLS counts in the NC/Nga mice notably increased from 126.3 ± 27.6 counts/24 h to 196.3 ± 23.0 and 331.3 ± 27.8 counts/24 h at 1 and 3 days, respectively, after IL-31 pretreatment (Figure 7a). The LLS counts in the BALB/c mice gradually increased from 101.8 ± 28.6 counts/24 h to 147.5 ± 27.0 and 222.5 ± 26.5 counts/24 h at 1 and 3 days, respectively, after IL-31 pretreatment (Figure 7b). The LLS counts in the C57BL/6 mice rapidly increased from 64.3 ± 10.4 counts/24 h to 182.1 ± 24.3 and 127.5 ± 13.4 counts/24 h at 1, and 3 days, respectively, after IL-31 pretreatment (Figure 7c). The IL-31RA-deficient (IL-31RA knock-in mouse: IL-31RAKI) mice showed no significant changes in LLS counts, which were 69.3 ± 13.6, 80.8 ± 14.0, and 61.8 ± 12.8 counts/24 h at 0, 1, and 3 days, respectively, after IL-31 pretreatment, (Figure 7d). The LLS counts significantly differed between the NC/Nga, BALB/c, C57BL/6, and IL-31RAKI mice.

### 2.8. Effects of Repeated Administration of IL-31 on IL-31RA Expression and Cohousing with Skin-Lesioned NC/Nga Mice-Induced Dermatitis Score in NC/Nga, BALB/c, and C57BL/6 Mice

The expression ratios of IL-31RA/β-actin in the NC/Nga, BALB/c, and C57BL/6 mice after the repeated administration of vehicle (PBS, 10 mL/kg, s.c., every 12 h for 3 days) and IL-31 (50 μg/kg, s.c., every 12 h for 3 days) were 4.68 ± 0.25, 2.45 ± 1.75, and 1.75 ± 0.27, respectively (Figure 8a). IL-31RA mRNA expression was significantly upregulated in the NC/Nga and BALB/c mice but not in the C57BL/6 mice. Dermatitis developed in the SPF-NC/Nga mice 4 weeks after cohousing. Dermatitis was also observed in the NC/Nga mice, and the symptoms gradually increased in severity. The dermatitis severity reached a maximum score of approximately 10.0 and plateaued at 10 weeks after cohousing (Figure 8b, blue line). Dermatitis developed in the BALB/c mice at 16 weeks after cohousing, and its symptoms gradually increased in severity, reaching a maximum score of approximately 4.0, which then plateaued 20 weeks after cohousing (Figure 8b, red line). The dermatitis score reached a steady state with approximately half the values of NC/Nga mice. The C57BL/6 mice did not develop dermatitis during the experimental period (Figure 8c, green line). Definite differences in dermatitis scores were found between the NC/Nga, BALB/c, and C57BL/6 mice.

## 3. Discussion

Itching is a characteristic symptom in various forms of dermatitis, especially AD; consequently, it constitutes a major diagnostic criterion [28]. Itch-associated scratching aggravates skin lesions, but the back does not develop dermatitis because it is not accessible to the patient’s hand and is, therefore, difficult to scratch [29]. NC/Nga mice with spontaneous skin lesions frequently scratch their face, ear, and rostral part of the back using their hind paws [30], which are accessible by the hind leg (Figure 1a, inside the red circle). Similar to humans, dermatitis also does not develop in the lower part of the mouse body as it is not accessible to their hind legs. All toenails of the NC/Nga mice were cut to inhibit toenail scratching, and the LLS counts did not change throughout the experimental period. Then, the dermatitis scores of the NC/Nga mice drastically decreased 1 week after cutting off the hind toenails, indicating that LLS by the toenails is an important factor in the dermatitis onset of NC/Nga mice [29].

Itch can be produced by a gentle touch, pressure, vibration, thermal, and electrical stimuli, such as transcutaneous or direct nerve stimulation. While itching may have different causes, it feels the same to our senses. Moreover, current studies on itching are based on the human pruriceptive sense, and these nociceptive stimuli have no discernible differences. However, the sensory perception of a foreign substance and true itching could be differentiated by dividing the scratching behavior of mice into LLS (itch-associated scratching) and SLS (hygiene behavior) [3]. Based on our evaluation standard, histamine is not a pruritogen. The i.d. injection of histamine can only trigger SLS. We think that the evaluation by subjectivity is absolute, but an itch and a tickling feeling cannot be distinguished definitely in practice. It is the sense that itching strips off the epidermis and removes an alien substance. Histamine causes cutaneous edema but does not cause the stripping of the epidermis, even with repeated i.d. injections of histamine to the skin.

Cohousing with skin-lesioned NC/Nga mice increases LLS counts and triggers dermatitis. In addition, several strains of mice, including SPF-NC/Nga, BALB/c, ICR, WBB6F1-W/Wv (W/Wv mice, mast cell-deficient), and C.B.17/Icr-scid (Scid mouse, T/B cell-deficient mice), show increased scratching behavior after a few days of cohousing with skin-lesioned NC/Nga mice. This finding suggests that serum IgE levels, T/B cells, and mast cells are unnecessary for the development of itch-associated scratching [5]. Moreover, direct stimulation by the movement of mites from the skin of one mouse to that of another may induce scratching. These results indicate that mite parasitism and not allergic reaction is involved in the increase in LLS counts and the development of dermatitis after cohousing with skin-lesioned NC/Nga mice. Moreover, cutaneous IL-31 mRNA expression is upregulated in mice cohoused with skin-lesioned NC/Nga mice [31].

The scratching behavior caused by IL-31 is distinct from that caused by other pruritogens (i.e., histamine, serotonin, compound 48/80, substance P, and PAR2). A single i.d. injection of IL-31 elicits LLS but not SLS, with the LLS beginning gradually approximately 1 h after injection and persisting over 24 h. In contrast, histamine [3], serotonin [32,33], compound 48/80 [34] substance P [10], or PAR2 [33] elicits SLS but not LLS, with the SLS beginning immediately after the i.d. injection and persisting for at least 30 min. In the present study, we administered IL-31 using different methods, such as subcutaneous, intradermal, intravenous, intracerebroventricular, and intraperitoneal, among which the intravenous injection caused the earliest and strongest induction of LLS. Preliminary test results showed no significant differences in the results of the other administration methods. These results indicated that IL-31 reached the site of action of IL-31RA after IL-31 entered the blood circulation. Given its similar effect in mite infestation, the i.d. route of injection of IL-31 was selected in subsequent experiments to examine LLS development. Meanwhile, given its similar effects in the administration of pruritogens and algogens, the intravenous injection of IL-31 was used to examine the induction of alloknesis because intravenous injections caused the earliest and strongest induction of LLS.

The itch-scratch cycle promotes further itching and aggravates skin lesions in patients with AD [30], possibly explaining why scratching aggravates itching and induces a vicious cycle of scratching-induced itching [35]. Despite the major therapeutic implications of elucidating the precise mechanisms underlying itchy skin, a suitable animal model for this phenomenon is lacking. “Itchy skin” is defined as an area of skin within which mechanical stimuli that normally evoke only a sensation of touch produce an itch. In a previous study, we measured the LLS counts 30 min before and after mechanical scratching in skin-lesioned NC/Nga mice. We found no significant overall changes but discovered that the LLS count increases immediately after the application of mechanical scratching [21]. This result suggests that alloknesis occurs in skin-lesioned NC/Nga mice [22]. Cohousing with skin-lesioned NC/Nga mice increased the LLS counts with a circadian rhythm, i.e., the LLS counts clearly increased at nighttime but not at daytime (Figure 1d). IL-31-induced LLS also showed the same circadian rhythm. The SLS counts (hygiene behavior) in the same mice also showed a clear circadian rhythm. These results suggest that the mechanical stimulation caused by SLS of the back skin becomes an itch stimulation involved in itching, alloknesis, and high LLS counts [31].

In the present study, we investigated itching in response to several pruritogens or algogens in BALB/c mice after being cohoused with skin-lesioned NC/Nga mice or intravenous injections of IL-31 and an alloknesis-like response. The LLS counts particularly increased in the mice infested with mites or administered IL-31. These mice were sensitive not only to chemical but also several mechanical (e.g., epidermal skin stripped with a finger, needle pinching of the skin, or liquid infusion to skin) stimuli. These results suggested that IL-31 did not exert a pruritogen-like effect and caused alloknesis, where various non-itch stimuli became itch stimuli and induced itching. Pruritogens and algogens should originally cause notably different behavior in mice [36]. However, both stimulants showed LLS after mite infestation or IL-31 injection in mice. Moreover, the i.d. injection of strong pain stimulator bradykinin and capsaicin induced an itch-scratch cycle [22], where scratching led to more itching. Clinical studies showed that serotonin and bradykinin, as classic endogenous algogens, can become potent histamine-independent pruritogens in the lesioned skin of patients with AD [37,38], suggesting that the itch-scratch cycle is caused by alloknesis.

The expression level of cutaneous IL-31 gene transcripts significantly increased in the NC/Nga mice, and these changes coincided with increases in scratching behavior [31]. Therefore, the anti-IL-31 antibody reduced scratching behavior in NC/Nga mice with dermatitis [39]. However, the expression of cutaneous IL-31 and IL-31RA mRNA is temporary, unstable, and unpredictable [31]. Hence, we suggest that IL-31 is an important factor in inducing itching and promoting scratching in NC/Nga mice. By contrast, the expression of DRG IL-31RA is very stable. In the present study, repeated intermittent i.d. administration of IL-31 gradually increased the LLS counts and DRG neuronal IL-31RA expression. Similarly, cohousing with skin-lesioned NC/Nga mice caused mite infestation-induced LLS and increased DRG neuronal IL-31RA expression. Repeated administration of IL-31 also increased the expression of DRG neuronal OSMR but not that of DRG neuronal LIFR or gp130 mRNA [17]. These results suggest that IL-31 upregulates the expression of IL-31 heterodimeric receptors. Cutaneous IL-31 upregulates IL-31RA expression in the DRG neuron cell bodies, and IL-31-induced LLS is enhanced by DRG IL-31RA expression [31]. Therefore, we suggest that DRG neuronal IL-31RA is more suitable than IL-31 as a target for therapeutic treatment for AD. IL-31 and DRG neuronal IL-31RA are important factors inducing itching and promoting scratching and dermatitis. These factors are involved in the development of pruritic skin diseases, including AD, but the mechanisms underlying AD development in humans remain unclear. In fact, the highest expression of IL-31RA mRNA was found in the DRG [24]. Cevikbas et al. suggested that neuronal IL-31RA could serve as a target to manage TH2-mediated itch in diseases including AD [40].

Recently, we reported that capsaicin suppresses LLS and SLS for more than 72 h [41]. At this point, the expression of DRG IL-31RA mRNA is significantly decreased, whereas that of cutaneous IL-31RA shows no significant change. Thus, capsaicin suppresses LLS by inhibiting IL-31RA mRNA expression in the DRG [41]. However, IL-31RA expression inhibitors, such as tacrolimus, along with TRPV1 stimulation, are difficult to use as therapeutic drugs because a patient with AD experiences severe pain immediately after its application [42]. Capsaicin has been widely used as a reliable research tool for inducing pain, which is mediated by the action of TRPV1 in nociceptive afferent terminals [43]. The pain that is partially induced through the activation of TRPV1 by capsaicin is possibly responsible for suppressing itching sensations. An IL-31RA expression inhibitor in the DRG could be applied as a suitable therapeutic drug for AD. An immunosuppressive drug has been recently approved and is in use for AD treatment. However, the results of this study suggest that AD should be regarded as a functional disorder of the sensory nerve rather than a peripheral immune disease, and AD therapeutic drug development focusing on sensory nerve activity improvement is necessary for the future.

Increased scratching counts were closely correlated only with DRG neuronal IL-31RA mRNA expression in NC/Nga and BALB/c mice. These results suggest that cohousing-induced LLS regulated by neuronal DRG IL-31RA expression in several mouse strains without the involvement of immune-related cells. Repeated administration of IL-31 gradually increased LLS counts and IL-31RA mRNA expression in the DRG but not cutaneous IL-31RA mRNA expression [31]. Expression of cutaneous or DRG neuronal IL-31 mRNA was not detected during the experimental period. Mite infestation and IL-31 administration increased the LLS counts and upregulated DRG neuronal IL-31RA expression, and the LLS counts showed a close correlation with DRG neuronal IL-31RA expression in the NC/Nga and BALB/c mice. Increasing LLS counts and DRG IL-31RA expression in the BALB/c was less than half that in the NC/Nga mice. Then, the severity dermatitis score induced by cohousing with skin-lesioned NC/Nga mice in the BALB/c mice was less than half of that in the NC/Nga mice. In other words, the DRG neuronal IL-31RA expression, LLS counts, and dermatitis score showed parallel characteristics. Mite infestation and repeated IL-31 administration did not increase LLS counts and DRG IL-31RA expression in the C57BL/6 mice. Furthermore, repeated administration of IL-31 did not increase the LLS counts in the IL-31RAKI mice. IL-31RAKI mice were generated using C57BL/6 as a genetic background. The C57BL/6 mice reacted to IL-31 but did not show an upregulation of DRG IL-31RA. IL-31 administration upregulated the DRG IL-31RA expression and increased the LLS counts in the NC/Nga and BALB/c mice but not in the C57BL/6 mice. The dermatitis scores differed between the NC/Nga, BALB/c, and C57BL/6 mice. This study shows a relationship between IL-31, alloknesis, DRG neuronal IL-31RA expression, neuronal IL-31RA, LLS counts, and dermatitis severity. In addition, the results of this study indicate that neuronal IL-31RA expression in the DRG was the most important factor determining dermatitis severity. In other words, one may or may not develop AD even when exposed to the same environment. The individuality of AD development possibly depends on differences in DRG neuronal IL-31RA expression.

## 4. Materials and Methods

### 4.1. Animals

Male SPF-NC/Nga and skin-lesioned-NC/Nga mice aged 6–13 weeks and BALB/c and C57BL/6 mice aged 6–8 weeks were purchased from SLC Japan (Shizuoka, Japan). The generation of mice lacking IL-31RA (IL-31RA^−/−^) has been described previously [44].

The IL-31RAKI (IL-31RA-deficient) mice used in this study were on a C57BL/6 genetic background and were obtained from hybrid mutant mice originally created on a 129 SVJ-C57BL/6 background by backcrossing breeding over 15 generations. In this study, we used male homozygous (IL-31RAKI, IL-31RA^−/−^) and wild-type (IL-31RA^+/+^) mice matched for age. The animals were housed under a controlled temperature (23 ± 3 °C), humidity (50 ± 5%), and lighting (lights on from 7:00 am to 7:00 pm). All animals were provided free access to food and tap water. All procedures for animal experiments were approved by the Committee for Animal Experimentation at the International University of Health and Welfare in accordance with the Guidelines for Proper Conduct of Animal Experiments (Science Council of Japan, Tokyo, Japan, 2006).

### 4.2. Reagent

The mouse IL-31 cDNA spanning amino acids 24–163 of IL-31 was cloned in a frame with pET30A (Novagen, Darmstadt, Germany), and the construct was transformed into BL-21 cells (Novagen, Darmstadt, Germany). After induction with isopropyl-β-D-thiogalactopyranoside, IL-31 protein was purified under denaturation conditions by nickel-chelating sepharose (Qiagen, Benelux B.V., The Netherlands) and then dialyzed in PBS [45]. IL-31, at a dose of 50 μg/kg, was used for subcutaneous or intravenous injection, as previously described [31]. Histamine (Wako Jyunyaku, Osaka, Japan), serotonin (Sigma-Aldrich, St Louis, MO, USA), compound 48/80 (Sigma-Aldrich), acetylcholine (Wako Jyunyaku), bradykinin (Wako Jyunyaku), or capsaicin (Wako Jyunyaku) was dissolved in saline and intradermally administered on the back and neck of the mice.

### 4.3. Measurement of Scratching Counts

Scratching counts were measured as previously described [46]. A small magnet (1.0 mm diameter, 3.0 mm length) was implanted subcutaneously into both hind paws of isoflurane-anesthetized mice 24 h before the measurement. Each mouse was placed in an observation chamber (11 cm diameter, 18 cm height) surrounded by a circular coil, through which electric current, induced by movement of the magnets attached to the hind paws, was amplified and recorded. The number of spontaneous scratches was automatically detected and objectively evaluated via a computer using MicroAct (Neuroscience, Tokyo, Japan) [3,47]. The analysis parameters for detecting waves were as follows: threshold, 0.1 V; event gap, 0.2 s; minimum duration, 1.5 s; maximum frequency, 20 Hz; and minimum frequency, 2 Hz.

### 4.4. Evaluation of Dermatitis Scores

Dermatitis was evaluated by assigning an inflammation score. Inflammation of the face, ears, and anterior part of the body was scored as follows: 0, none; 1, mild; 2, moderate; and 3, severe. This scoring was based on the severity of erythema/hemorrhage, edema, excoriation/erosion, and scaling/dryness. The total score (minimum = 0, maximum = 12) for each mouse was taken as the score for that mouse [48].

### 4.5. Induction of Itching and Dermatitis

LLS was induced in SPF-NC/Nga, BALB/c, and C57BL/6 mice by cohousing them with skin-lesioned-NC/Nga mice for 1–14 days. Scratch count was measured as described above and compared between the groups. Dermatitis was induced in SPF-NC/Nga, BALB/c, and C57BL/6 mice by cohousing them with skin-lesioned-NC/Nga mice for 1–36 weeks. Each value represents the median ± quartile deviation.

### 4.6. Measurement of Compound-Induced Scratch Counts

The increase in LLS showed a circadian rhythm; there was a remarkable increase at night but not during daytime, particularly from 9:00 to 14:00. Therefore, we examined the scratching behavior induced by several stimulants from 9:00 to 14:00 [35]. To elicit scratching behavior, we injected 0.02 mL of histamine (10 μg), serotonin (2.5 μg), compound 48/80 (5 μg), acetylcholine (10 μg), bradykinin (10 μg), or capsaicin (1 μg) 1 h after the administration of IL-31 (50 μg/kg, intravenous). Immediately after the injection, the mice were placed in an observation chamber, and their scratching behavior was monitored every 10 min for 30 min.

### 4.7. Quantitative Real-Time PCR

The expression of IL-31, IL-31RA, and β-actin was measured using real-time polymerase chain reaction (RT-PCR) in the DRG (C_4–7_, T_1–4_) neuron cell body from the shoulder and back of the NC/Nga, BALB/c, and C57BL/6 mice at each point. Total RNA was extracted from the dorsal skin of each mouse by Trizol (Invitrogen, Carlsbad, CA, USA) and digested using amplification-grade DNase I (Invitrogen), according to the manufacturer’s instruction. cDNA was synthesized by the SuperScript III First-Strand Synthesis System (Invitrogen). Quantitative RT-PCR was performed with SYBR Green Master Mix, using an Applied Biosystems 7700 Sequence Detection System (Applied Biosystems, Foster City, CA, USA). The PCR primers for IL-31 were designed using Primer 3 software, and primers for β-actin were purchased from TAKARA BIO (Otsu, Shiga, Japan). Primer sequences were as follows: IL-31 (5′-ATA CAG CTG CCG TGT TTC AG-3′ and 5′-AGC CAT CTT ATC ACC CAA GAA-3′), IL-31RA (5′-CCA GAA GCT GCC ATG TCG AA-3′ and 5′-TCT CCA ACT CGG TGT CCC AAC-3′), and β-actin (5′-TGA CAG GAT GCA GAA GGA GA-3′ and 5′-GCT GGA AGG TGG ACA GTG AG-3′). Relative expression levels were calculated using the relative standard curve method as outlined in the manufacturer’s technical bulletin. A standard curve was generated using the fluorescence data obtained from four-fold serial dilutions of the total RNA of the sample with the highest expression. The curve was then used to calculate the relative amounts of target mRNA in the test samples. The quantities of all targets in the test samples were normalized to the corresponding β-actin RNA transcript levels in skin samples.

### 4.8. Statistical Analysis

All data were statistically analyzed using GraphPad InStat and GraphPad Prism (GraphPad Software Inc., La Lolla, CA, USA). Experimental values are expressed as means and standard errors (SE). Data on time-course changes in scratching counts or the percentage of mRNA expression were analyzed using a two-way analysis of variance (ANOVA) followed by the Bonferroni test. One-way ANOVA followed by the Student–Newman–Keuls multiple comparison test was used to compare other data. The correlation between time-course changes in the percentage of mRNA expression and scratching counts induced by mite infestation or IL-31 administration at each time after cohousing with skin-lesioned NC/Nga mice or administration of IL-31 was investigated, and the product-moment correlation coefficient was calculated. *p*-values less than 0.05 were considered statistically significant.

## 5. Conclusions

AD is a common skin disease caused by genetic and environmental factors. In this study, we examined the genetic factors contributing to the onset of itch-associated scratching in different strains of mice. We compared the effects of mite infestation or IL-31 administration on the LLS counts and dermatitis scores of NC/Nga, BALB/c, and C57BL/6 mice. The results showed that mite-infestation and IL-31 administration increased the LLS counts and DRG neuronal IL-31RA mRNA expression and dermatitis scores in the NC/Nga and BALB/c mice but not in the C57BL/6 mice. This result can be ascribed to the differences in IL-31-induced DRG neuronal IL-31RA expression between the mice. Overall, the results of this study suggest that the neuronal IL-31RA expression in the DRG is the most important genetic factor in mice. In other words, one may or may not develop AD even when exposed to the same environment. The individuality of AD development possibly depends on differences in DRG neuronal IL-31RA expression.

## Figures and Tables

**Figure 1 ijms-24-01047-f001:**
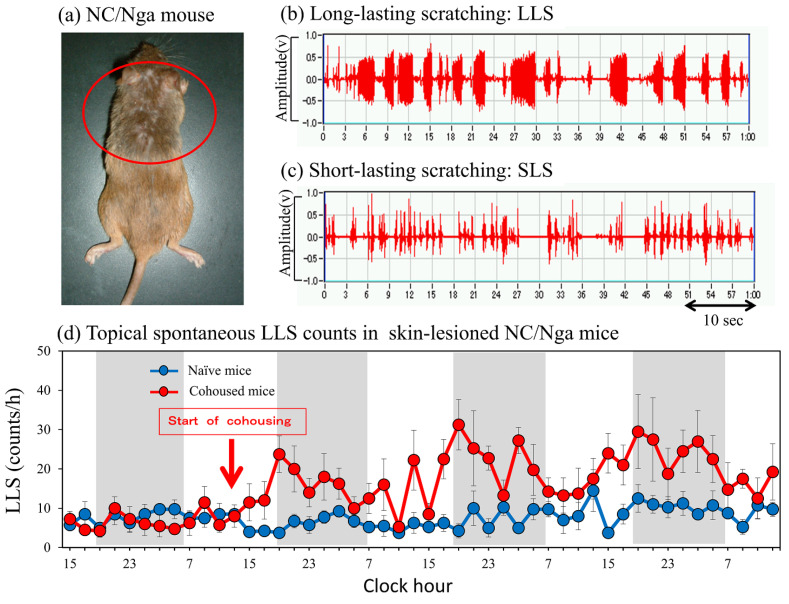
Scratching behavior of skin-lesioned NC/Nga mice. (**a**) Skin−lesioned NC/Nga mice, an animal model of atopic dermatitis. NC/Nga mice with spontaneous skin lesions frequently scratch their faces, ears, and rostral parts of their backs (inside the red circle) using their hind paws. Dermatitis also does not develop in the lower part of the mouse body, which is not accessible to its hind paws. (**b**) Representative form of long-lasting scratching (LLS, scratching behavior that lasts more than 1.5 s, itch-associated scratching behavior). (**c**) Representative form of short-lasting scratching (SLS, scratching behavior that lasts from 0.3 to 1.5 s, associated hygiene behavior). (**d**) LSL counts of SPF-NC/Nga mice cohoused with skin-lesioned NC/Nga mice. Red arrow indicates the start of cohousing with skin-lesioned NC/Nga mice, the lateral axis indicates the clock hour, and the shaded area represents nighttime (dark phase, 7:00 p.m. to 7:00 a.m.).

**Figure 2 ijms-24-01047-f002:**
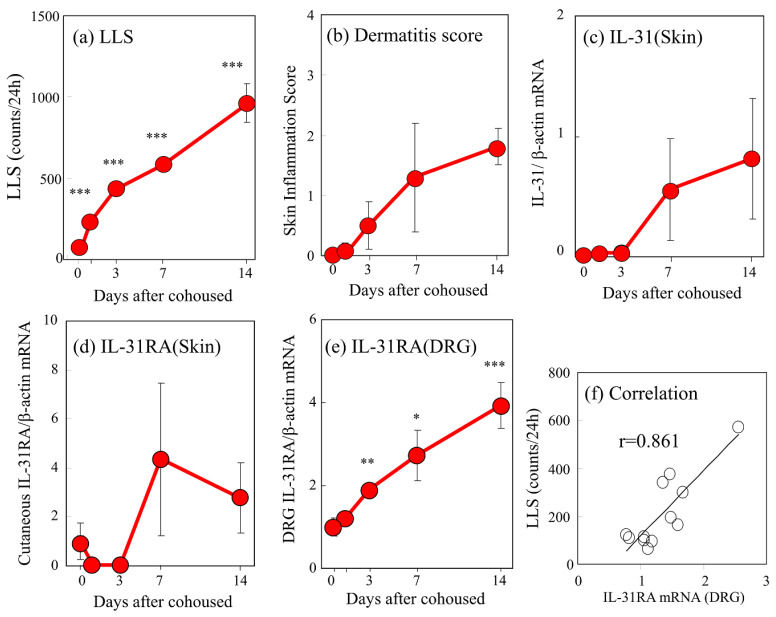
Time-course changes in long-lasting scratching (LLS), dermatitis score, cutaneous interleukin-31 (IL-31) and IL-31 receptor A (IL-31RA) mRNA expression, and dorsal root ganglion (DRG) neuronal IL-31RA mRNA expression in SPF-NC/Nga cohoused with skin-lesioned NC/Nga mice. (**a**) LLS (over 1.5 s) in NC/Nga mice was measured after 1–14 days. (**b**) Dermatitis score was based on the severity of erythema/hemorrhage, edema, excoriation/erosion, and scaling/dryness. (**c**) Cutaneous IL-31 mRNA/β-actin mRNA expression ratio. (**d**) Cutaneous IL-31RA mRNA/β-actin mRNA ratio. (**e**) DRG neuronal IL-31RA mRNA/β-actin mRNA expression ratio. (**f**) Correlation between the DRG neuronal IL-31RA mRNA expression ratio and the LLS counts in NC/Nga mice. Neuronal IL-31RA mRNA/β-actin mRNA expression ratio in SPF-NC/Nga mice cohoused with skin-lesioned NC/Nga mice for 3, 7. and 14 days was measured as described in the text. The values represent the means ± standard error (S.E.) from six mice. * *p* < 0.05, ** *p* < 0.01, *** *p* < 0.001 compared with pre-cohoused values of SPF-NC/Nga mice (Student’s *t*-test with Bonferroni correction).

**Figure 3 ijms-24-01047-f003:**
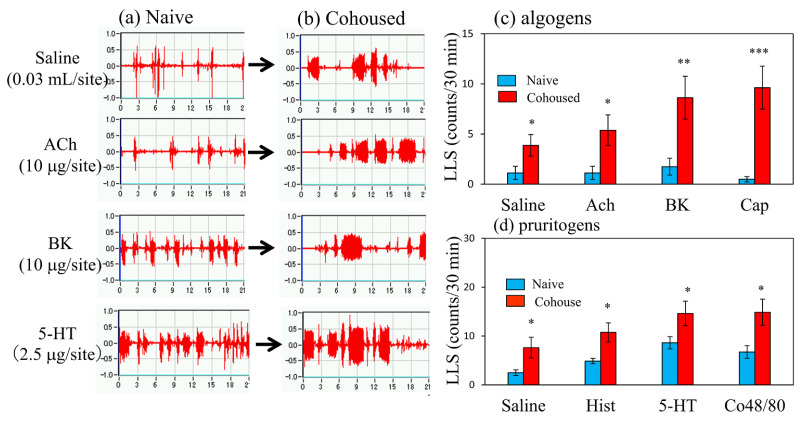
Effects of several algogens or pruritogens on long-lasting scratching (LLS) counts in BALB/c mice cohoused with skin-lesioned NC/Nga mice for 6 days. (**a**) Typical traces of saline-, acetylcholine-, bradykinin-, and serotonin-induced scratching behavior in BALB/c mice with itchy skin induced by cohousing (mite infestation). Traces show the scratching behavior of naïve mice. (**b**) Traces show the scratching behavior of itchy skin induced by mite infestation (**c**) Effects of itchy skin induced by mite infestation on LLS counts induced by saline (0.03 μL/site, intradermal i.d.), acetylcholine (ACh, 10 μg/site, i.d.), bradykinin (BK, 10 μg/site, i.d.), or capsaicin (Cap, 1 μg/site) in mice. (**d**) Effects of itchy skin induced by mite infestation on scratching counts induced by saline (0.03 mL/site, i.d.), histamine (Hist, 10 μg/site, i.d.), serotonin (5-HT, 2.5 μg/site, i.d.), or compound 48/80 (Co48/80, 10 μg/site) in mice. Each value represents the mean ± standard error (S.E.) from six mice (total of 24 mice). * *p* < 0.05, ** *p* < 0.01, and *** *p* < 0.001 compared with the respective values in naïve mice (Student’s *t*-test).

**Figure 4 ijms-24-01047-f004:**
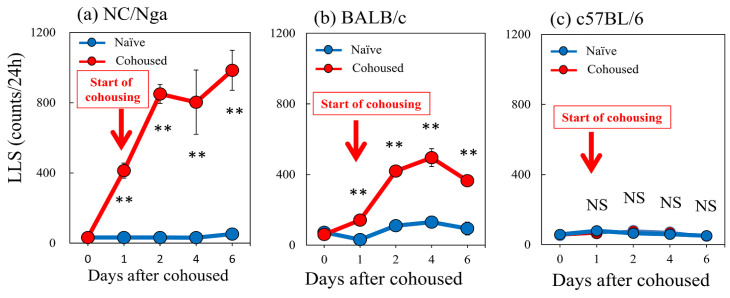
Time-course changes in long-lasting scratching (LLS) counts caused by mite infestation between NC/Nga, BALB/c, and C57BL/6 mice. (**a**) Mite infestation increased LLS counts in SPF-NC/Nga mice. (**b**) Mite infestation increased LLS counts in BALB/c mice. (**c**) In C57BL/6 mice, no significant changes in LLS counts were observed. Blue line indicates naïve mice. Red lines indicate cohoused mite-infested mice. Red arrow indicates the start of cohousing with skin-lesioned NC/Nga mice. Each value represents the mean ± standard error (S.E.) from six mice (total of 36 mice). NS, not significant. ** *p* < 0.01 when compared with the respective value of naïve mice (Student’s *t*-test with Bonferroni collection).

**Figure 5 ijms-24-01047-f005:**
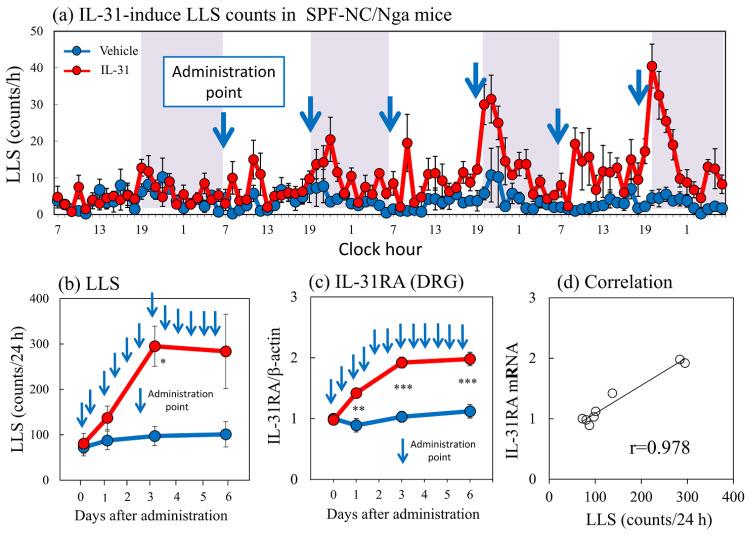
Effect of repeated administration of IL-31 on long-lasting scratching (LLS) counts and dorsal root ganglion (DRG) neuronal IL-31RA mRNA expression in SPF-NC/Nga mice. (**a**) LLS counts induced by repeated intradermal (i.d.) administration of IL-31 in NC/Nga mice. Repeated injection of IL-31 (1 μg/site, i.d.) every 12 h for 3 days gradually increased LLS counts, which increased intermittently after each dosage of IL-31. This increase in LLS counts showed a circadian rhythm; in particular, the LLS counts significantly increased at nighttime. Blue arrows indicate the administration points of vehicle or IL-31. Lateral axis indicates the clock hour, and the shaded area represents nighttime (7:00 pm to 7:00 am). (**b**) Effect of repeated administration of IL-31 on LLS counts for 6 days. (**c**) Effect of repeated administration of IL-31 on IL-31RA mRNA expression in the DRG. (**d**) Correlation between LLS counts and IL-31RA mRNA expression in the DRG (r = 0.978). Blue lines indicate vehicle (phosphate buffer solution, PBS)-administered mice, and red lines indicate IL-31 (1 μg/site, i.d., every 12 h for 3 days)-administered mice. Each value represents the mean ± standard error (S.E.) from six mice. NS, not significant. * *p* < 0.05, ** *p* < 0.01, and *** *p* < 0.001 compared with the respective values in the vehicle-administered group (Student’s *t*-test with Bonferroni collection).

**Figure 6 ijms-24-01047-f006:**
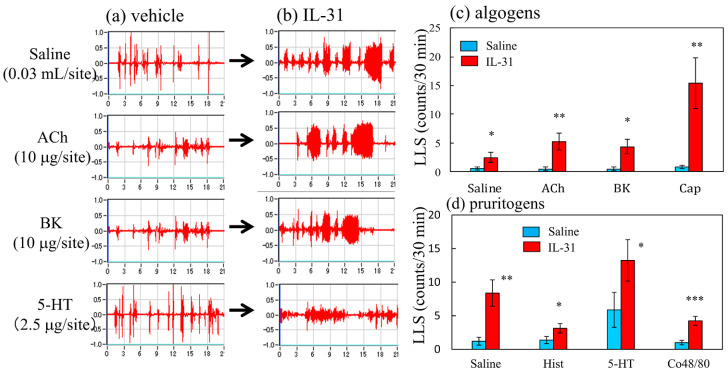
Effects of several algogens or pruritogens on long-lasting scratching (LLS) counts 1 h after the intravenous injection of IL-31. (**a**) Typical traces of saline-, acetylcholine-, bradykinin-, and serotonin-induced scratching behavior in vehicle (PBS, 10 mL/kg, i.v.)- or IL-31-administered (50 μg/kg, i.v.) BALB/c mice. Traces show the scratching behavior of the vehicle-administered mice. (**b**) Traces on the right side show the scratching behavior of IL-31-administered BALB/c mice. (**c**) Effects of IL-31 on mice treated with acetylcholine (ACh, 10 μg/site, intradermal i.d.), bradykinin (BK, 10 μg/site, i.d.), or capsaicin (Cap, 1 μg/site, i.d.). (**d**) Effects of IL-31 on scratching counts induced by saline (0.03 mL/site, i.d.), histamine (Hist, 10 μg/site, i.d.), serotonin (5-HT, 2.5 μg/site, i.d.) or compound 48/80 (Co48/80, 10 μg/site) in mice. Each value represents the mean ± standard error (S.E.) from six mice (total of 24 mice). * *p* < 0.05, ** *p* < 0.01, and *** *p* < 0.001 compared with the respective values in the vehicle-administered group (Student’s *t*-test).

**Figure 7 ijms-24-01047-f007:**
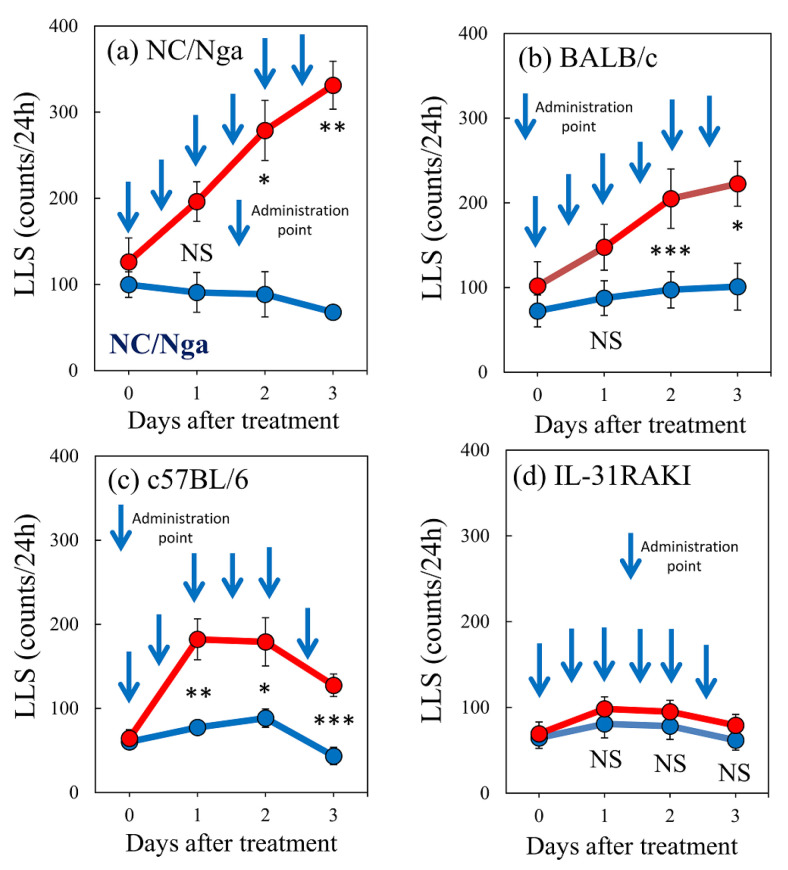
Effect of repeated administration of IL-31 on long-lasting scratching (LLS) counts between NC/Nga, BALB/c, C57BL/6, and IL-31RAKI mice. Effect of repeated administration of IL-31 on the LLS counts of (**a**) SPF-NC/Nga mice. (**b**) BALB/c, (**c**) C57BL/6, and (**d**) IL-31RAKI (IL-31RA-deficient) mice. Blue arrows indicate the vehicle (phosphate-buffered saline (PBS), 10 mL/kg, subcutaneous, every 12 h for 3 days) or IL-31 (50 μg/kg, s.c., every 12 h for 3 days) administration points. Blue lines indicate vehicle-administered mice, and red lines indicate IL-31-administered mice. Each value represents the mean ± standard error (S.E.) from six mice (total of 48 mice). NS. not significant. * *p* < 0.05, ** *p* < 0.01, and *** *p* < 0.001 when compared with the respective value of vehicle-treated mice (Student’s *t*-test with Bonferroni collection).

**Figure 8 ijms-24-01047-f008:**
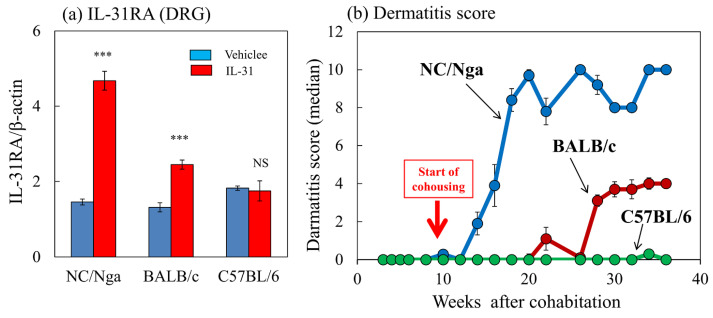
Comparison of IL-31-induced dorsal root ganglion (DRG) IL-31RA expression and dermatitis development between NC/Nga, BALB/c, and C57BL/6 mice. (**a**) Effect of repeated administration of IL-31 on IL-31RA mRNA expression in the DRG in NC/Nga, BALB/c, and C57BL/6 mice. Blue columns indicate vehicle (phosphate-buffered saline (PBS), 10 mL/kg, subcutaneous (s.c.), every 12 h for 3 days)-administered mice, and red columns indicate IL-31 (50 μg/kg, s.c., every 12 h for 3 days)-administered mice. (**b**) Differences in the development of dermatitis caused by cohousing with skin-lesioned NC/Nga mice (mite infestation) between SPF-NC/Nga, BALB/c, and C57BL/6 mice. Red arrow indicates the start of cohousing with skin-lesioned NC/Nga mice. Each value represents the mean ± standard error (S.E.) from a total of six mice. NS. not significant. *** *p* < 0.001 when compared with the respective value of vehicle-treated mice (Student’s *t*-test).

## Data Availability

Not applicable.

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
