# Peer review of "Interleukin-31 Receptor A Expression in the Dorsal Root Ganglion of Mice with Atopic Dermatitis"

_ijms, 2023, doi:10.3390/ijms24021047_

Round 1
Reviewer 1 Report
In this study by Arai et al. , the authors have found that the augmented expression of Interleukin-31 R in the DRG is crucial to induce the itch associated with atopic dermatitis. Although results are convincing, I find the organization of the article very hard to follow. Given below are my comments to improve the manuscript.
· “The number of spontaneous scratches is automatically detected and objectively evaluated by a computer [3]”. This line can be moved to the methods because it breaks the continuity of the introduction.
· “We found that LLS 34 (over 1.5 s) is frequently observed in skin-lesioned NC/Nga mice, whereas SLS (0.3–1.5 s) 35 is frequently observed in skin-lesioned and not skin-lesioned NC/Nga and other strains of mice.”This is a part of the results that needs to be moved to the relevant section. Or cite a reference if it is a part of already published results.
· “ These results suggest that SLS is a social and hygiene behavior (locomotor ac-37 tivity), whereas LLS is the true itching response in these mice.” The authors write as SLS itch as a social behavior. Please delete it as itch is not a social behavior.
· “we hypothesized that IL-31 is an important factor inducing itching and promoting scratching in NC/Nga mice.” After citing relevant literature on the role of IL-31 in itch and AD, the authors write about their hypothesis. The previous citations (lines 44-54) show that most of the work with IL-31 has been already done. Authors need to re-write the hypothesis and add it after they introduce IL-31R
· Please put Il-31 and IL-31R part of the introduction together. Delete 64-67. Please condense the information for Il-31 and IL-31R, there is lot of redundancy and repeated information which does not go well with a reader
· “This finding is interesting because 86 the sensation of itch, similar to pain, is directly mediated by unmyelinated C fibers of 87 primary sensory neurons, whose cell bodies reside within the DRG [25].”
Delete this information, first of all, it is irrelevant to the flow of the introduction. Second of all and more importantly, the statement is not true. A-fibers also mediate some types of pain and mechanical itch.
· Please do not oversay “This is the first article that shows a relationship between 95 IL-31, alloknesis, DRG neuronal IL-31RA expression, neuronal IL-31RA, LLS counts, and 96 dermatitis severity”. Delete this the first study.
· “2.3. Effects of Pruritogens and Algogens on LLS Counts in BALB/c Mice Cohoused with Skin-134 Lesioned NC/Nga mice” This should come after “2.4. Effects of Cohousing with Skin-Lesioned NC/Nga Mice on LLS Counts in NC/Nga, BALB/c, 161 and C57BL/6 Mice.”
· Why c57BL/6 did not develop itch on being co-housed with mite-infested mice? In the legend the authors 4C write “Mite-infestation increased LLS counts in C57BL/6 mice” which is not what the results show. Figure 3 the legibility of the font in the figure is very low. Please increase the resolution of the figure image.
· “The LLS counts significantly differed between 171 the NC/Nga, BALB/c, and C57BL/6 mice.” No value of the statement. Please delete.
· Line 192-193 “cutaneous …experiment” This statement is ambiguous and does not convey what authors wants to say
· Line 218 Correct typo “ At h”
· Discussion needs to be shortened and the redundancy along with irrelevant information should be removed.
Author Response
Response to Reviewer 1
Point 1: “The number of spontaneous scratches is automatically detected and objectively evaluated by a computer [3]”. This line can be moved to the methods because it breaks the continuity of the introduction.
Response 1: Thank you for pointing this out. We agree with this suggestion, This sentence has been moved to the methods section, accordingly. Lines 485-487, The number of spontaneous scratches was automatically detected and objectively evaluated via a computer using MicroAct (Neuroscience, Tokyo, Japan) [3,47].
Point 2:· “We found that LLS 34 (over 1.5 s) is frequently observed in skin-lesioned NC/Nga mice, whereas SLS (0.3–1.5 s) 35 is frequently observed in skin-lesioned and not skin-lesioned NC/Nga and other strains of mice.”This is a part of the results that needs to be moved to the relevant section. Or cite a reference if it is a part of already published results.
Response 2: Thank you for your valuable suggestion. As this content a part of the previously published results, we have included reference [3,4] in Line 36.
Point 3:· “ These results suggest that SLS is a social and hygiene behavior (locomotor activity), whereas LLS is the true itching response in these mice.” The authors write as SLS itch as a social behavior. Please delete it as itch is not a social behavior.
Response 3: Thank you for your thorough review of our manuscript. We apologize for this error. We have deleted the term “social” from the revised text (Lines 37, 68-69, 317, and 364-365).
Point 4:· “we hypothesized that IL-31 is an important factor inducing itching and promoting scratching in NC/Nga mice.” After citing relevant literature on the role of IL-31 in itch and AD, the authors write about their hypothesis. The previous citations (lines 44-54) show that most of the work with IL-31 has been already done. Authors need to re-write the hypothesis and add it after they introduce IL-31R
Response 4: Thank you for this suggestion, We have revised our study objectives.
Lines, 57-58. Our recent finding showed that IL-31 is an important factor inducing itching and promoting scratching in NC/Nga mice; however, the sites of action of IL-31 remain unclear.
Point 5:· Please put Il-31 and IL-31R part of the introduction together. Delete 64-67. Please condense the information for Il-31 and IL-31R, there is lot of redundancy and repeated information which does not go well with a reader
Response 5: Thank you for your valuable suggestion. However, a phenomenon called allonesis is not generally known. Because we want to leave this part as why we examined an alloknesis-like phenomenon with a mice lose a reason when alloknesis does not explain in introduction section.
Point 6:· “This finding is interesting because 86 the sensation of itch, similar to pain, is directly mediated by unmyelinated C fibers of 87 primary sensory neurons, whose cell bodies reside within the DRG [25].” Delete this information, first of all, it is irrelevant to the flow of the introduction. Second of all and more importantly, the statement is not true. A-fibers also mediate some types of pain and mechanical itch.
Response 6: Thank you for this suggestion. The statement has been deleted from the revised manuscript.
Point 7:· Please do not oversay “This is the first article that shows a relationship between 95 IL-31, alloknesis, DRG neuronal IL-31RA expression, neuronal IL-31RA, LLS counts, and 96 dermatitis severity”. Delete this the first study.
Response 7: Thank you; this suggestion has been complied with.
Point 8:· “2.3. Effects of Pruritogens and Algogens on LLS Counts in BALB/c Mice Cohoused with Skin-134 Lesioned NC/Nga mice” This should come after “2.4. Effects of Cohousing with Skin-Lesioned NC/Nga Mice on LLS Counts in NC/Nga, BALB/c, and C57BL/6 Mice.”
Response 8: Expression of LLS includes circadian rhythm and hardly develops in the day. Therefore, I think that there is little influence of LLS of the autogenesis because I carry out the experiment during a period from 9:00 to 14:00.
We added the sentences as follows, Lines 504-506. This increase in LLS showed a circadian rhythm: while there was a clear increase at night, the increase during daytime was not so clear, particularly from 9:00 to 14:00 (Fig. 1B). Therefore, we examined the scratching behavior induced by several stimulants from 9:00 to 14:00.
Point 9:·Why c57BL/6 did not develop itch on being co-housed with mite-infested mice? In the legend the authors 4C write “Mite-infestation increased LLS counts in C57BL/6 mice” which is not what the results show. Figure 3 the legibility of the font in the figure is very low. Please increase the resolution of the figure image.
Response 9: We do not know the cause that why C57BL/6 did not develop LLS caused by cohousing with skin-lesioned NC/Nga mice.
As hard copy did the screen of the computer as for Figure 3. and Figure 6., resolution is inferior. We think that you can understand difference between LLS and SLS.
Point 10:· “The LLS counts significantly differed between 171 the NC/Nga, BALB/c, and C57BL/6 mice.” No value of the statement. Please delete.
Response 10: In accordance with your suggestion, we have deleted this sentence.
- Line 192-193 “cutaneous …experiment” This statement is ambiguous and does not convey what authors wants to say
I apologize for this error. It is cutaneous IL-31 mRNA and IL-31RA mRNA expression. I corrected We wanted to show that the expression of cutaneous IL-31 and IL-31RA mRNA did not influence by exogenous administration of IL-31.We revised the sentence as follows. Lines, 192-193. Cutaneous IL-31 and IL-31RA mRNA were not expressed during the experiment.
Point 11: Line 218 Correct typo “ At h”
Response 11: Thank you for your thorough review. We have corrected this error. Line, 228.
Point 12: Discussion needs to be shortened and the redundancy along with irrelevant information should be removed.
Response 12: We agree with the reviewer’s suggestion, we deleted the redundant sentences as follows,
However, these phenomena can be blocked by insecticidal pretreatment, suggesting that mite-infestation is responsible for the increased LLS counts and dermatitis [7].
The onset of itching related gene expression is impracticable due to the need to resect DRG neuron cell bodies from a living human. These results suggest that AD should be regarded as a functional disorder of the sensory nerve rather than a peripheral immune disease.
And Conclusion section
Close correlations were found between LLS counts and DRG neuronal IL-31RA mRNA expression. Mite-infestation and IL-31 administration also increased dermatitis severity in the NC/Nga and BALB/c mice but not in the C57BL/6 mice. The dermatitis scores of the BALB/c mice were approximately half those of the NC/Nga mice, and their LLS counts and dermatitis scores showed parallel characteristics. The dermatitis scores significantly differed between the NC/Nga, BALB/c, and C57BL/6 mice.
Thank you for your consideration of our manuscript.
Reviewer 2 Report
The manuscript entitled "Interleukin-13 receptor A expression in the dorsal root ganglion of mice with atopic dermatitis” by Arai and Saito is an excellent piece of work and hence suitable for publication in the International Journal of Molecular Sciences. In this manuscript, the authors conducted an extensive investigation using a range of experiments. It is very interesting to study the effects of repeated administration 14 of IL-31 on the scratching behavior in NC/Nga, BALB/c, and C57BL/6 mice. The results also well documented and discussed accordingly. However, before to accept for publication, an authors needs to be addressed the following comments
Comments to authors
1. Line 16: what is LLS is not clear, need to disclose the full form
2. Line 21: what is IL-31RA is not clear. Need to clarify to readers
3. The details of the figures should be in the caption. In figure 1 author mentioned the details at each subfigure itself. it is recommended to put all information as a caption
4. Subheading 2.2. is too long, the authors needs to concise it as much as possible
5. Figure 2, 2d at day 7, the standard deviation is too high. Can the author explain the reason or re-conduct the study to minimize the errors if any
6. Figure 5a, look at comment no.3.
7. Line 200. Why does the sentence start with (a). or if it is part of the caption, there should not be a space/empty line
8. The above comment is also applicable to lines 225 and 226.
9. Conclusion missing the major outcome of the study. Need to be taken care of
Author Response
Response to Reviewer 2
Point 1: Line 16: what is LLS is not clear, need to disclose the full form
Response 1: Thank you for this suggestion. We added the following sentence. Lins 32-33, long-lasting scratching (LLS: scratching behavior lasting more than 1.5 s) and short-lasting scratching (SLS: scratching behavior lasting from 0.3 and 1.5 s).
Point 2: Line 21: what is IL-31RA is not clear. Need to clarify to readers
Response 2: We have added the following sentences. Lines 8–11:
Atopic dermatitis (AD) is a common skin disease caused by hereditary and environmental factors. However, the mechanisms underlying AD development remain unclear. In this study, we examined the hereditary factors contributing to the onset of itch-associated scratching in different strains of mice.
Point 3: Lines 21-22, IL-31RA expression in the DRG is the most important genetic factor affecting the severity of dermatitis in mice.
Pesponse 3: The details of the figures should be in the caption. In figure 1 author mentioned the details at each subfigure itself. it is recommended to put all information as a caption
Response 3: We agree with the suggestion. Accordingly, we added the following sentence in the caption of Figure 1. Lines 62-69. (a) Skin-lesioned NC/Nga mice, an animal model of atopic dermatitis. NC/Nga mice with spontaneous skin lesions frequently scratch their faces, ears, and rostral parts of their backs (inside the red circle) using their hind paws. Dermatitis also does not develop in the lower part of the mouse body, which is not accessible to its hind paws. (b) Representative form of long-lasting scratching (LLS, scratching behavior that lasts more than 1.5 s, itch-associated scratching behavior). (c) Representative form of short-lasting scratching (SLS, scratching behavior that lasts from 0.3 to 1.5 s, associated with locomotor activity and/or hygiene behavior).
Point 4: Subheading 2.2. is too long, the authors needs to concise it as much as possible
Response 4: We changed Subheading 2.2 as follows. Lines, 115-116: 2.2. Time-Course Changes in Several Parameters of Dermatopathy in SPF-NC/Nga Mice Cohoused with Skin-Lesioned NC/Nga Mice
Point 5: Figure 2, 2d at day 7, the standard deviation is too high. Can the author explain the reason or re-conduct the study to minimize the errors if any
Response 5: Probably it is suggested that I repeat expression and disappearance by the situation of the tick infection intermittently.
Point 6: Figure 5a, look at comment no.3.
Response 6: We have added the following sentences. Lines 206-210, (a) LLS counts induced by repeated intradermal (i.d.) administration of IL-31 in NC/Nga mice. Blue arrows indicate the administration points of vehicle or IL-31. Repeated intradermal injection of IL-31 (1 mg/kg, i.d.) every 12 h for 3 days gradually increased LLS counts; it increased at every dosage of IL-31 intermittently. This increase in LLS counts showed a circadian rhythm; in particular, the LLS counts significantly increased at night-time. Lateral axis indicates the clock hour, and the shaded area represents nighttime (7:00 pm to 7:00 am).
Point 7: Line 200. Why does the sentence start with (a). or if it is part of the caption, there should not be a space/empty line
Response 7: We thank for your indication. We removed the empty lines/spaces in all figure captions.
Point 8: The above comment is also applicable to lines 225 and 226.
Response 8: As above, this has been addressed.
Point 9: Conclusion missing the major outcome of the study. Need to be taken care of
Response 9: Thank you for this valuable suggestion. We added the following sentences. Lines 546-548,
AD is a common skin disease caused by genetic and environmental factors. In this study, we examined the hereditary factors contributing to the onset of itch-associated scratching in difference strains of mice.
Lines 554-555, Overall, the results of this study suggest that the neuronal IL-31RA expression in the DRG is the most important genetic factor in mice.
Thank you for your consideration of our manuscript.
Round 2
Reviewer 1 Report
Authors put SLS as locomotor activity, please delete this as these words are unclear and do not clearly communicate what the authors want to say.
Author Response
Response to Reviewer 1
Point 1: Authors put SLS as locomotor activity, please delete this as these words are unclear and do not clearly communicate what the authors want to say.
Response 1: Thank you for your thorough review of our manuscript. We apologize for this error. We have deleted the term “locomotor activity” from the revised text (Lines 37, 68, 317, and 365).
Thank you for your consideration of our manuscript.
Regards,
Iwao Arai, Ph.D.,
Division of Environmental Allergy,
The Jikei University School of Medicine
